# Investigation of the Structural Strength of Existing Blast Walls in Well-Test Areas on Drillships

**Byeongkwon Jung [1], Jeong Hwan Kim [2] and Jung Kwan Seo [1,2,***

[1] Department of Naval Architecture and Ocean Engineering, Pusan National University, Busan 46241, Korea; bk47.jung@gmail.com

[2] The Korea Ship and Offshore Research Institute, Pusan National University, Busan 46241, Korea; kjhwan1120@pusan.ac.kr

* Correspondence: seojk@pusan.ac.kr; Tel.: +82-51-510-2415

**Abstract:** Blast walls are installed on the topside of offshore structures to reduce the damage from fire and explosion accidents. The blast walls on production platforms such as floating production storage, offloading, and floating production units undergo fire and explosion risk analysis, but information about blast walls on the well-test area of drillship topsides is insufficient even though well tests are performed 30 to 45 times per year. Moreover, current industrial practices of design method are used as simplified elastically design approaches. Therefore, this study investigates the strength characteristic of blast wall on drillship based on the blast load profile from fire and explosion risk analysis results, as well as the ability of the current design scantling of the blast wall to endure the blast pressure during the well test. The maximum plastic strain of the FE results occurs at the bottom connection between the vertical girder and the blast wall plate. Based on the results, several alternative design applications are suggested to reduce the fabrication cost of a blast wall such as differences of stiffened plated structure and corrugated panels, possibility of changing material (mild steel), and reduced plate thickness for application in current industrial practices.

**Keywords:** drillship; blast wall; explosion; well-test areas; strength evaluation; non-linear structural analysis

## 1. Introduction

One of the most important trends in the development of the modern oil and gas industry is reorientation to the development of offshore oil and gas fields. The first offshore drilling rigs were constructed in 1934 on drilling shelves with stationary substructures [1]. The development of continental shelves has resulted in a variety of accidents with catastrophic consequences due to the lack of attention devoted to the identification of prevention and mitigation of possible risks and hazard.

The most serious accidents on drilling ships and platforms of various types (semi-submersible, submersible, mobile, stationary) between 1979 and 2015 are included in historical databases [2]. Well-known examples include "Piper Alpha" in the North Sea, "Deepwater Horizon" in the Gulf of Mexico and "SOCAR (The State Oil Company of the Azerbaijan Republic)" in the Caspian Sea. These accidents resulted from flooding, explosion and fire, and oil slippages. The distribution of accidents shows that a considerable proportion of accidents are due to personnel errors, such as process disturbance, improper pilotage, and improper berthing to offshore oil and gas facilities. According to statistical data, these emergency incidents are distributed by Fattakhova and Barakhnina [3]. Incidents such as fire, explosion and oil slippages occurred mostly during drilling work.

Drillships have the functional ability of semi-submersible drilling rigs but greater mobility (i.e., they are ship-shaped vessels), and they can move quickly under their own propulsion from drill site to another drill site, in contrast to semi-submersibles, jack-up barges, and platforms. A generally

drillship comprises a hull, mud module, subsea control module, mud process module, drill floor, pipe and casing storage, and well-test module. One unusual characteristic is that the hull bottom has a large opening (so called "moon-pool") that allows the seabed to be drilled with a drill pipe and casing. A typical drillship has a length of 220 to 230 m, a breadth 36 to 42 m and a depth of 18 to 20 m, and drilling equipment on its topside. Its main advantage is its ability to perform drilling operations in very deep sea (up to 12 km), where general fixed platforms are not able to be drilled.

Among a drillship's various operations, the well-test operation is similar to a process module of a production offshore platform. Therefore, the safety design of a drillship should consider fires and explosions. For example, water-based drilling fluids (WBDF), which if used with proper additives such as nano-particles can be more effective than conventional oil-based drilling fluids and also can mitigate the firing problems for high pressure and high-temperature wells. Oil-based drilling fluids can be toxic to the marine species, if any incident of oil spills happens, such as the Gulf of Mexico, whereas, considered WBDF with nano-particles are non-toxic [4–6].

For those reasons, the well-test area on drillship installed blast walls for minimizing and preventing explosion damage surrounding modules. The structural design of the blast wall should consider the type of explosive load to which the structure will be exposed because of the numerous uncertainties inherent in an explosion. According to this reason, numerous investigations of blast walls on offshore platforms have been conducted to determine design methods and for structural evaluation though theoretical, numerical, and experimental approaches.

The literature includes many studies on the use of corrugated and stiffened walls on offshore platforms. A formal framework was suggested for blast walls with FE analysis [7]. That study provided not only a better understanding of the underlying assumptions required to justify blast wall explosive load detection behavior but also a coherent basis for specification and design for dynamic analysis of blast wall capabilities. In the presence of theoretical studies of blast walls, they proposed a detailed and reasonable FE model and conducted two analyses: traditional non-linear static analysis and non-linear time-path dynamic analysis. Vignjevic et al. [8] considered different internal reinforcements (C-section and corrugated) to improve the energy absorption properties of thin walled rectangular beams under uniaxial and biaxial deep bending collapse, for loading angles. Also, corrugated reinforcements showed a greater potential for increasing specific energy absorption, which was supported by investigating key geometric parameters, including corrugation angle, depth, and number using LS-DYNA [9] simulations experimentally validated. Sohn et al. [10] studied the role of a flat-plated stiffener on the structural characteristics of a blast wall on an offshore installation exposed to hydrocarbon explosions. Blast walls are generally installed in oil and gas production structures to minimize and prevent the damage from explosions. Kang et al. [11] suggested a blast loading application method. The uniformly distributed loading condition, predicted by explosion risk analysis, was applied in most previous analysis methods. However, the analysis methods related to load conditions are inaccurate because the blast overpressure around the wall tends to show a low level in open areas and a high level in enclosed areas. Syed et al. [12] suggested a method for analysis of offshore stainless-steel blast walls. Blast walls are mostly designed using simplified analytical techniques, such as the single degree of freedom method, in which global deformation or displacement is used as the primary response parameter. This study uses detailed non-linear Finite Element analysis to present realistic responses of offshore blast walls under various high impulsive pressure loads generated from accidental hydrocarbon explosions. The numerical models were also verified against past experiments results with similar steel blast panels. An extensive parametric study was conducted in which the verified FE models were used to construct pressure-impulse (P-I) diagrams for various deformation levels. Sohn and Kim [13] studied the structural response of corrugated blast walls depending on blast load pulse shapes. Hydrocarbon explosions are among the most hazardous events for workers on offshore platforms. Corrugated blast walls are typically installed to protect structures against explosion loads, but the profiles of real explosion loads differ depending on the congestion and confinement of the topside structures. Xiao et al. [14] suggested numerical prediction of blast

wall effectiveness for structural protection against air blasts. The propagation of shock waves and their interaction with the blast wall were simulated in FE code (LS-DYNA [9,15,16]) to calculate the overpressure-time profile of the blast wall. The influence of structural flexibility on the effectiveness of the blast wall was considered by including a steel sheet ("canopy") at the top of the wall. These studies help to understand the behavior of an air blast interacting with blast walls and helps to identify the principal parameters for the design of such walls.

According to existing blast wall studies are well developed, validated, and suggested. However, limited studies have examined the blast wall in the well-test area on the topside of a drillship, likely because although floating offshore platforms (e.g., floating liquid natural gas and floating production storage and offloading units) have great exposure to the risk of fire and explosion accidents, accidents aboard drillships are expected to occur during well test operation. Moreover, current industrial practices of blast wall design on drillships, design method is simplified elastically design approaches. The design of the blast wall around the well-test area was evaluated according to the bending and shear stress of the supporting vertical girder using a static pressure (0.2 bar) and beam theory, and this method of evaluation may be over-designed when considering the blast pressure against a realistic blast load. Furthermore, only beam theory is applied to evaluate the blast wall, so it was not possible to evaluate the connection of the vertical girder and deck. Therefore, a blast wall on a drillship should be re-evaluated with advanced and/or developed methods for proper and accurate design.

## 2. Blast Wall of Well-Test Area on Drillship Topside

### 2.1. Drillship Topside Arrangement and Well-Test Area

To determine the target drillship topside arrangement and well-test area, an existing drillship was considered for this study as shown in Figure 1. The blast wall is placed between the well-test area and the ROV (Remotely Operated Vehicle) station on the port side. In addition, the lifeboat stations on either side of the drillship are located further aft than the well-test area and are partially shielded by the funnel casings.

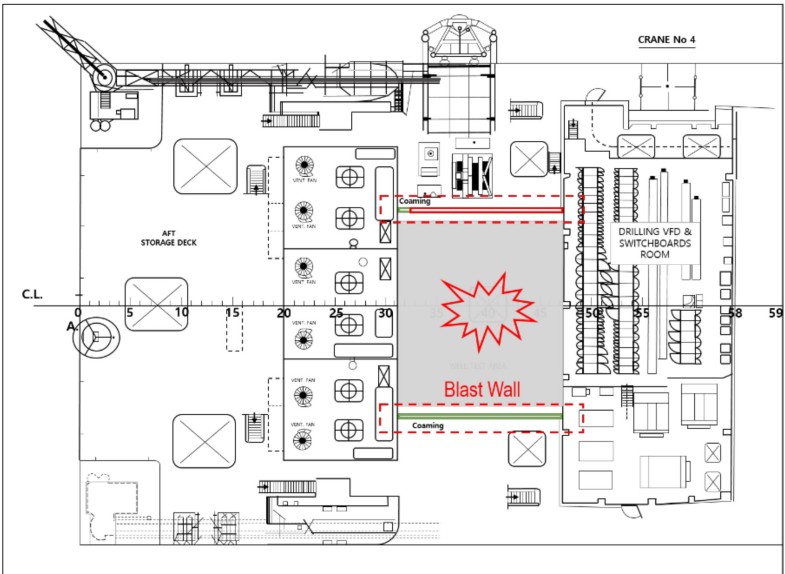

**Figure 1.** Typical well-test area arrangement on a drillship.

Well testing is normally performed approximately 30 to 45 times per year on a drillship [17]. The objectives of well testing are divided into to obtain and analyze representative samples of produced fluids, to measure the reservoir pressure and temperature, to determine the inflow performance

relationship and deliverability, to evaluate the completion efficiency, to characterize well damage, and to evaluate work over and simulation treatments.

During well-test operation, oil or gas leakage is likely to result in an explosion accident. Therefore, to protect the equipment and structures on the topside, a blast wall is installed beside the expected ignition point. Figure 2 shows that the blast wall on the drillship topside is a stiffened panel structure that differs from a general blast wall and/or corrugated structure.

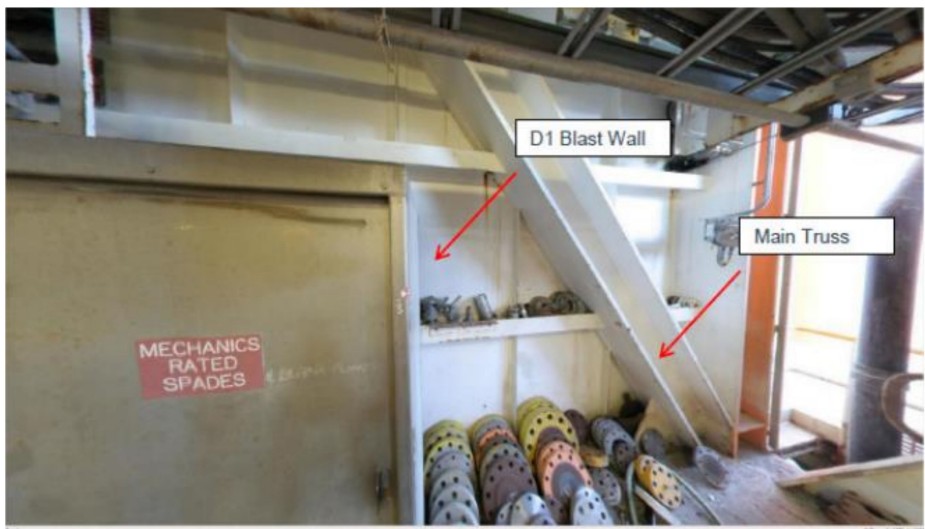

**Figure 2.** Example of a stiffened panel blast wall on drillship.

### 2.2. Characteristic of Explosion Load Profile

In industrial practice, simple beam theory has been applied to evaluate blast wall support with calculated blast pressure. The applied blast pressure is only considered static pressure without blast load profiles such as the duration time and the peak time.

In this study, to identify the load profile and scenarios of the well-test area, two specific studies were thoroughly reviewed. A previous experimental study [18] identified the characteristics of overpressure loads in an explosion, as shown in Figure 3a. A test module was fabricated for the explosion test that was similar to the processing module of a gas explosion in a floating liquid natural gas unit, and the explosion pressures were measured with a different geometrical effect, which is termed "porosity" and is given by Equation (1):

$$\text{Porosity} = 1 - \frac{\text{Volume of total structure}}{\text{Volume of total space}} \tag{1}$$

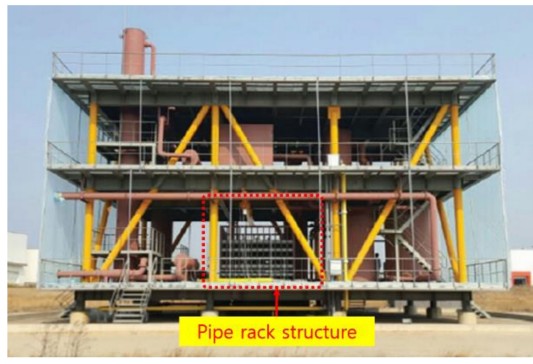

(**a**)

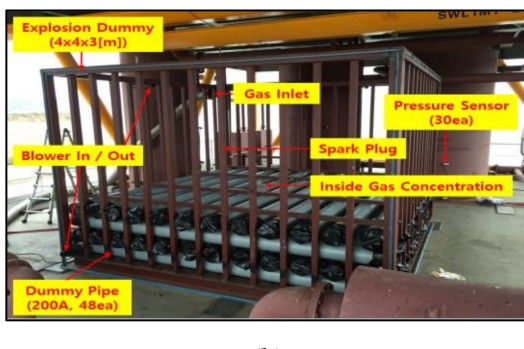

(**b**)

**Figure 3.** Full-scale test module for determination of explosion load profile. (**a**) Test module of a typical offshore topside. (**b**) Pipe rack structure for explosion.

The porosity was controlled by the number of pipes at the ignition point with a pipe rack structure for ignition, as shown in Figure 3b. Five scenarios were investigated with 0, 12, 24, 36, and 48 pipes. The ignition point was at the center of gas cloud.

As the experimental results [18] provide that the overpressure loads rose to a peak value within a very short period of rise time and decayed sharply in the experimental results. The rise time until the peak overpressure loads were reached differed depending on the considered structural congestion model. The rise time became shorter with increase in the degree of structural congestion. The overpressure loads fell into negative values compared with the ambient pressure and were recovered to the ambient pressure as the impact energy was released. In this result, porosity is an important factor in the overpressure in an explosion, and a structure with higher porosity has higher overpressure. The experimental results allow the explosion pressure profile to be categorized as "Arrival time", "Positive phase", and "Negative phase", respectively.

The general load profile of an explosion is quite complicated due to the explosion condition. The realistic characterization of blast pulse pressure action requires the pressure-time history to be traced, including the rise time, peak pressure, duration, and type of pressure decay. Blast pressure can be idealized as an impulsive loading that is characterized by peak pressure and duration time. To simplify the structural analysis, the time history of the panel load around its peak was idealized as a triangular impulse. The idealized blast load profiles can be placed into four categories, as shown in Figure 4. Those profiles were analyzed with three different domains: the impulsive domain ($\frac{\tau}{T} \leq 0.3$), the dynamic domain ($0.3 \leq \frac{\tau}{T} < 3$) and the quasi-static domain ($3 \leq \tau/T$). The structural behavior characteristics of corrugated blast walls and stiffened blast walls were simulated under various types of explosion loadings [5,8].

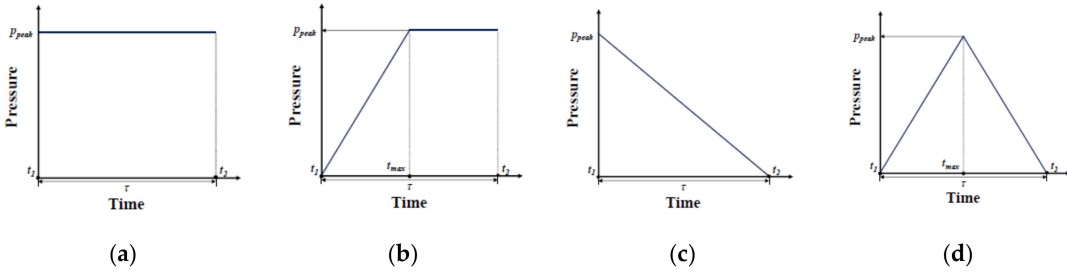

**Figure 4.** Common blast loading shapes [19]. (**a**) Rectangular. (**b**) Gradually applied. (**c**) Linear decay. (**d**) Triangular.

For selection of blast loading shapes, Sohn and Kim [13] provided that the effects of the shape of the load profile on the structural characteristic of blast walls show that the duration and peak pressure are not critical, but impulse is important in the impulsive domain. The rising time and the level of impulse have profound influences on the structural characteristics of blast walls in the dynamic domain. The results indicate that the structural characteristics of blast walls depend on the amount of applied momentum rather than the peak pressure or duration time. Furthermore, loads with linear decay and triangular loads have the same impulse value, but the ductility ratio differs greatly in the dynamic domain while remaining very similar in the impulse domain. In other words, the rising time is not critical in the impulse domain, but it is very important in the dynamic domain.

In fact, the current blast wall structure on drillship is designed with the simply linear beam theory approach, therefore the structure is assumed as a rigid body, and the stresses are concentrated at the bottom end of the vertical supports. According to these investigations, this study of blast load profile assumed a triangular load profile of positive phase and effect of negative phase.

## 2.3. Method of Strength Assessment of a Blast Wall in a Well-Test Area

An applicable method for the strength assessment of a blast wall in a well-test area suggested both fire/explosion analysis and structural analysis, as shown in Figure 5. Fire and explosion risk analysis is performed to evaluate accidental loading scenario hazards to define potential accidents that may occur to the facility during its lifetime from installation to decommissioning and assess the corresponding risk exposure. Potential accidents are defined as scenarios involving a ship collision, a dropped object, a fire, or a blast that introduces risk to personnel, the environment, or the facility.

To estimate the fire/explosion risk from the well-test area, certain operational days of well testing are assumed and included in the overall risk picture in the general oil/gas industrial risk practices. For advanced and quantitative risk evaluation, details of the equipment and piping and instrumentation diagrams for the well-test unit to be installed on the drill ship should be conducted with available date and calculation methods.

A typical well-test unit can be used to assess several components (valves, flanges, pie sections, etc.) for use as the basis of leak frequency assessment. Leak rates and frequencies are also calculated for each component, failure mode and type of medium with generic data or statistical methods.

The well-test unit has been divided into two segments such as gas and oil. Components that contain both oil and gas are allocated to the gas segment. This allocation is done due to the difficulties in predicting the amount of gas or oil that will be released during a leak (which depends on the gas/oil ratio in the reservoir to be tested). This is a conservative approach and is applied for components upstream of the separator from the drill floor, which is included in the gas segment.

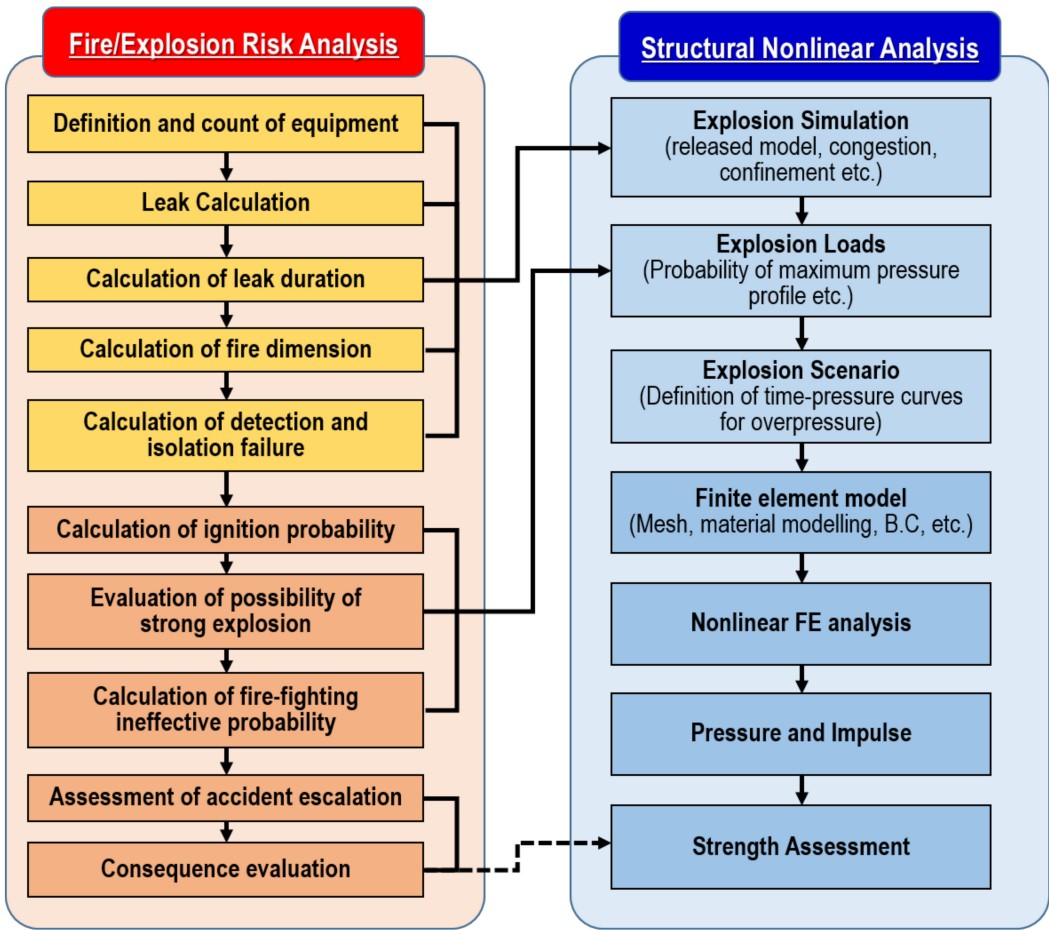

**Figure 5.** Flow chart risk and structural analyses of this study.

To evaluate the strength of the blast wall in this study, the explosion load requires for non-linear structural analysis. In addition, the design of the blast wall may be difficult because it requires predicting the type of explosive load to which a structure may be exposed due to the numerous uncertainties inherent to an explosion. Generally, oil/gas industries, classification societies and international/national organizations are used to idealized random explosion loads with symmetric triangular shapes because these well match an actual explosion loading with respect to attributes such as total impulse, peak pressure, rising time, and pulse duration. Therefore, the explosion load profile can be defined via fire/explosion risk analysis. Figure 5 (right side) shows the structural response characteristics of blast walls with various explosion loading shapes determined via non-linear dynamic FE analysis (procedure of structural non-linear analysis).

### 2.3.1. Calculation of Explosion Load Profile

The well-test area and adjacent areas have been assessed for potential fire and explosion accidents during well testing in current industrial practice [20]. The flow chart in Figure 5 presents the steps in the analysis. The ballast wall strength of the explosion load depends on factors such as the fuel type, congestion, confinement, and relative position of the ignition point within the gas cloud.

To determine the blast effect and prediction of explosion pressure, DNV-Phast [21] is used with the Baker–Strehlow–Tang model, which uses assumed parameters (material reactivity: low; flame expansion: 1.5; obstacle density: high; ground reflection factor: 2.0).

The affected volume ($30\,\mathrm{m}^3$) is considered the confined volume within the gas cloud. The maximum explosion load can be read at around 5.0 m from the gas cloud and the ignition point. According to the analysis result, the explosive pressure based on the leak category calculation is shown in Table 1.

The arrangement of the well test and adjacent area in Figure 6 shows that the blast wall in the well-test area is located approximately 10 m from the ignition point; therefore, the peak pressure on the blast wall can be assumed via linear interpolation with the pressure value 5 m from the ignition point and the distance of the peak pressure 0.2 bar. Various peak pressures were also selected to evaluate the blast wall for load scenarios (0.2, 0.3, 0.6, 1.0, 1.5, 2.0, 2.5 and 3.0 bar).

**Table 1.** Overview of major explosion accidents.

| Leak Category | Mass Used for Explosion (kg) | Maximum Explosion Load (Barg, at 5 m) | Distance to 0.2 Barg (m) | Peak Pressure on Blast Wall (Barg) |
|---|---|---|---|---|
| Small | - | - | - | - |
| Medium | 0.47 | 0.65 | 13 | 0.37 |
| Major | 8.44 | 3.0 | 33 | 2.5 |
| Large | 131.34 | 3.4 | 51 | 3.05 |

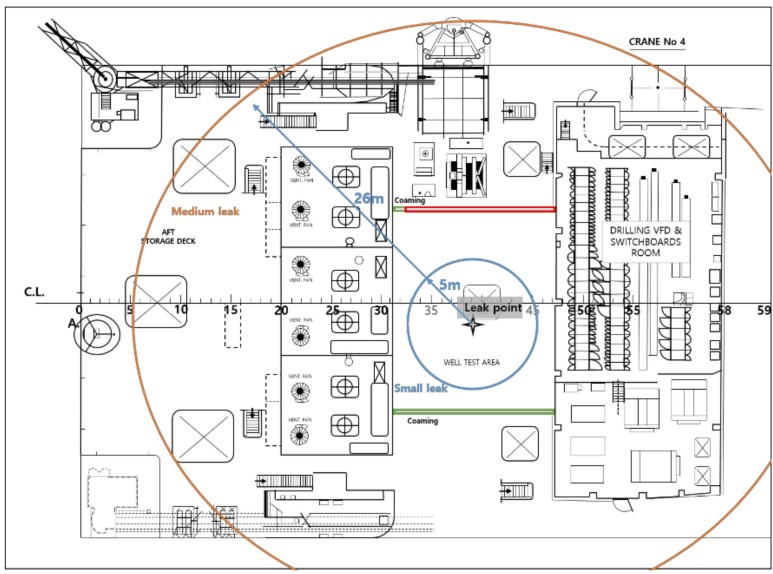

**Figure 6.** Arrangement of well test and adjacent area.

### 2.3.2. Applied Blast Load Scenarios

Various factors affect the duration time and thus the impulse of the blast pressure. The impulse is presented in Equation (2):

$$\text{Impulse (bar; s)} = \text{Peak pressure (bar)} \times \text{Duration time (s)} \tag{2}$$

The selected impulses are 0.001, 0.01, 0.02, 0.03, 0.04, 0.05 and 0.06 bar·s, and the duration time is calculated with Equation (2). Based on a review of the blast load profile study, the load applied in this study was a triangular loading pulse without the negative impulse, as shown in Figure 7. The considered load scenarios are presented with the selected peak pressures and durations in Table 2 and in Appendix A (Table A1. Load scenario of peak pressure at 0.3, 0.6, 1.0, 1.5, 2.0, 2.5 and 3.0 bar).

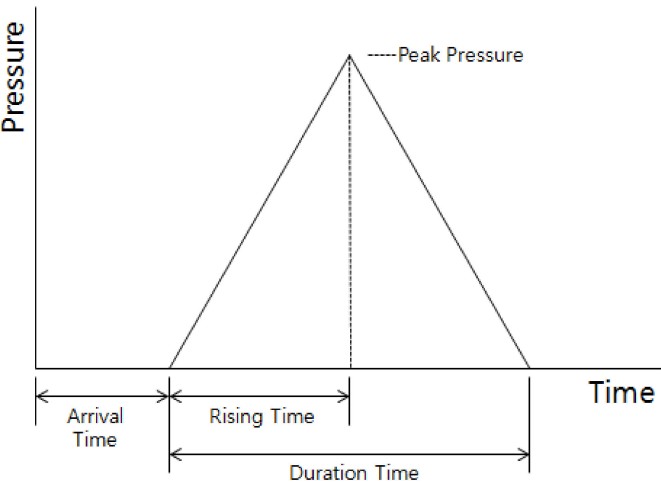

**Figure 7.** Idealized load profile in the analysis.

**Table 2.** Load scenario (0.2 bar).

| Peak Pressure (bar) | Case Number * | Duration Time (s) | Peak Time (s) | Impulse (bar) |
|---|---|---|---|---|
| | P02M01 | 0.0100 | 0.0050 | 0.001 |
| | P02M10 | 0.1000 | 0.0500 | 0.010 |
| | P02M20 | 0.2000 | 0.1000 | 0.020 |
| **0.2** | P02M30 | 0.3000 | 0.1500 | 0.030 |
| | P02M40 | 0.4000 | 0.2000 | 0.040 |
| | P02M50 | 0.5000 | 0.2500 | 0.050 |
| | P02M60 | 0.6000 | 0.3000 | 0.060 |

* Note: The letter P is peak pressure, and the next two numbers are the value of peak pressure without a decimal point (0.2 bar indicated as 02); the letter M is impulse, and next two numbers are the value of impulse without a decimal point (0.001 bar indicated as 01).

## 2.4. Numerical Modelling of Blast Wall

Target Blast Wall

In this study, the commercial non-linear FE code (MSC/Nastran [22]) was applied to simulate and analyze highly non-linear structural characteristics by considering geometrical and material nonlinearities. Iso-parametric quadrilateral elements with four nodes and six degrees of freedom per node were used to analyze thin to moderately thick shell structures such as the corrugated steel plate elements [23–25]. Figure 8 shows the FE model of the blast wall in the well-test area on the drillship topside. A normal blast wall has a corrugated shape, but the blast wall in the well-test area is a stiffened panel. Details of the dimensions of the stiffeners and material properties are given below:

- Blast wall thickness: 10 mm (AH36)
- Vertical H-beam: H400 × 400 × 13 × 24 (AH36)
- Horizontal H-beam: H200 × 200 × 10 × 15 (AH36)
- Horizontal channel: C200 × 100 × 7.5 × 11 (AH36)

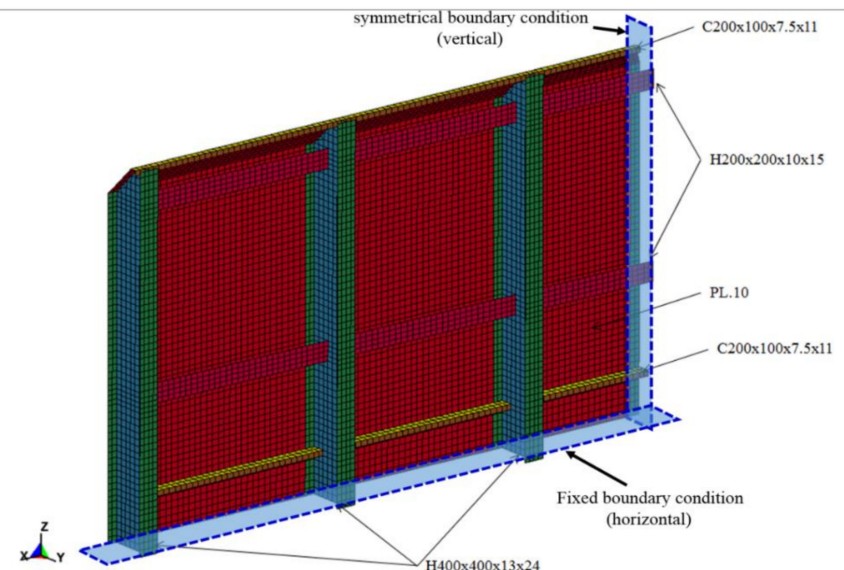

**Figure 8.** Scantling of blast wall and boundary conditions.

To represent the plastic behavior in a non-linear FE analysis, the material behavior obtained from the tensile tests [10]. Because elasto-perfectly plastic model was used for material model, only yield stress was converted to dynamic yield stress for FE simulations. In an impact load case such as explosion load, the effect of the strain-rate must be significant in the FE material model. The dynamic effect is the Cowper–Symonds equation, which follows as Equation (3), to consider the strain effect of material:

$$\frac{\sigma_{Yd}}{\sigma_Y} = 1.0 + \left(\frac{\dot{\varepsilon}}{C}\right)^{1/q} \tag{3}$$

where the coefficients C and q can be determined based on the test data [10]. The data indicate that the coefficients C and q are dependent on the material type, among other factors. The C and q values in this study were selected as 3200 and 5.0, respectively, for high-tensile steel and 40.4 and 5.0, respectively, for mild steel.

Figure 8 shows the boundary condition of the FE model. For the horizontal fixed condition, the vertical girders are welded on the deck and the blast wall edges are free, and for the vertical symmetrical boundary condition, the horizontal stiffeners and blast wall edge all have symmetrical boundary conditions ($\delta_x$ : free, $\delta_y$ : fixed, $\delta_z$ : free, $\gamma_x$ : fixed, $\gamma_y$ : free, $\gamma_z$ : fixed) for the continued installation condition.

## 2.5. Analysis Results

Figure 9 shows typical results for the distribution of a von Mises stress plot and plastic strain plot of case P15M20 (peak pressure: 1.5 bar; impulse: 0.02 bar·s) for the target blast wall structure and the selected load scenarios in the previous sections. The maximum von Mises stress is observed at the connection of the vertical H-beam and the deck at 0.13 s, with a duration of 0.0333 s. A low stress level is calculated, which may be explained by the rigidity of the vertical H-beam (H400 × 400 × 13 × 24), which takes most of the load from the explosion.

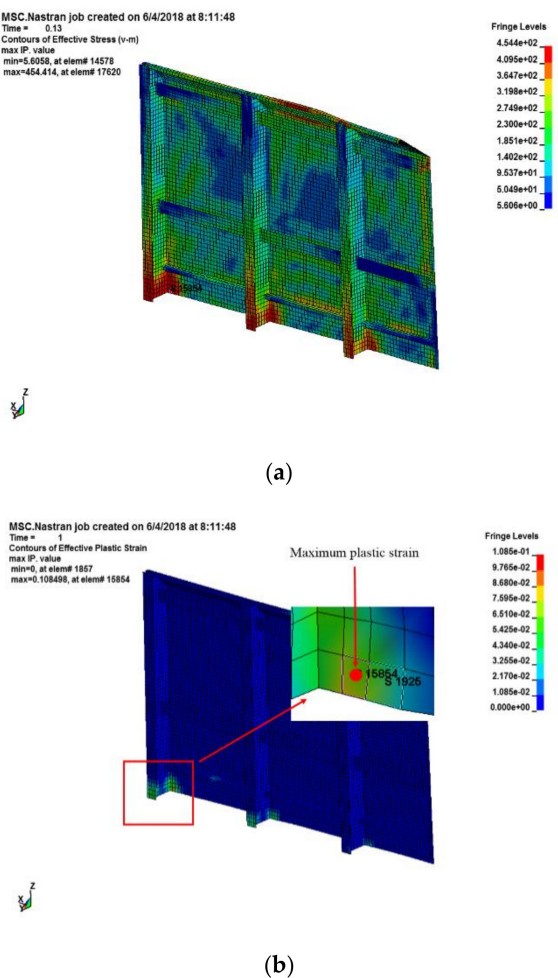

(**a**)

(**b**)

**Figure 9.** Stress and strain distribution of P15M20. (**a**)Von Mises stress. (**b**) Plastic strain.

Most of the blast wall plates do not reach the plastic strain limit compared with the three connections of the vertical H-beam and the deck. The furthest connection from the vertical symmetric boundary line has greater plastic strain than the other two connections. From this result, it can be concluded that the explosion load is concentrated on the three bottom connections (especially the connection furthest from the symmetric boundary) between the blast wall, and the deck for the blast wall has generally been designed with elastic criteria without consideration of energy absorption by deflection.

Figure 10a shows the relationship between plastic strain and the impulse curve of the blast wall at the maximum point. The plastic strain is increased when the peak pressure and the impulse are increased. The maximum and permanent deflection according to the peak pressure and impulse are summarized in Figure 10b,c. The maximum and permanent deflection have tendencies similar to the plastic strain, and the permanent deflection is slightly lower than the maximum deflection.

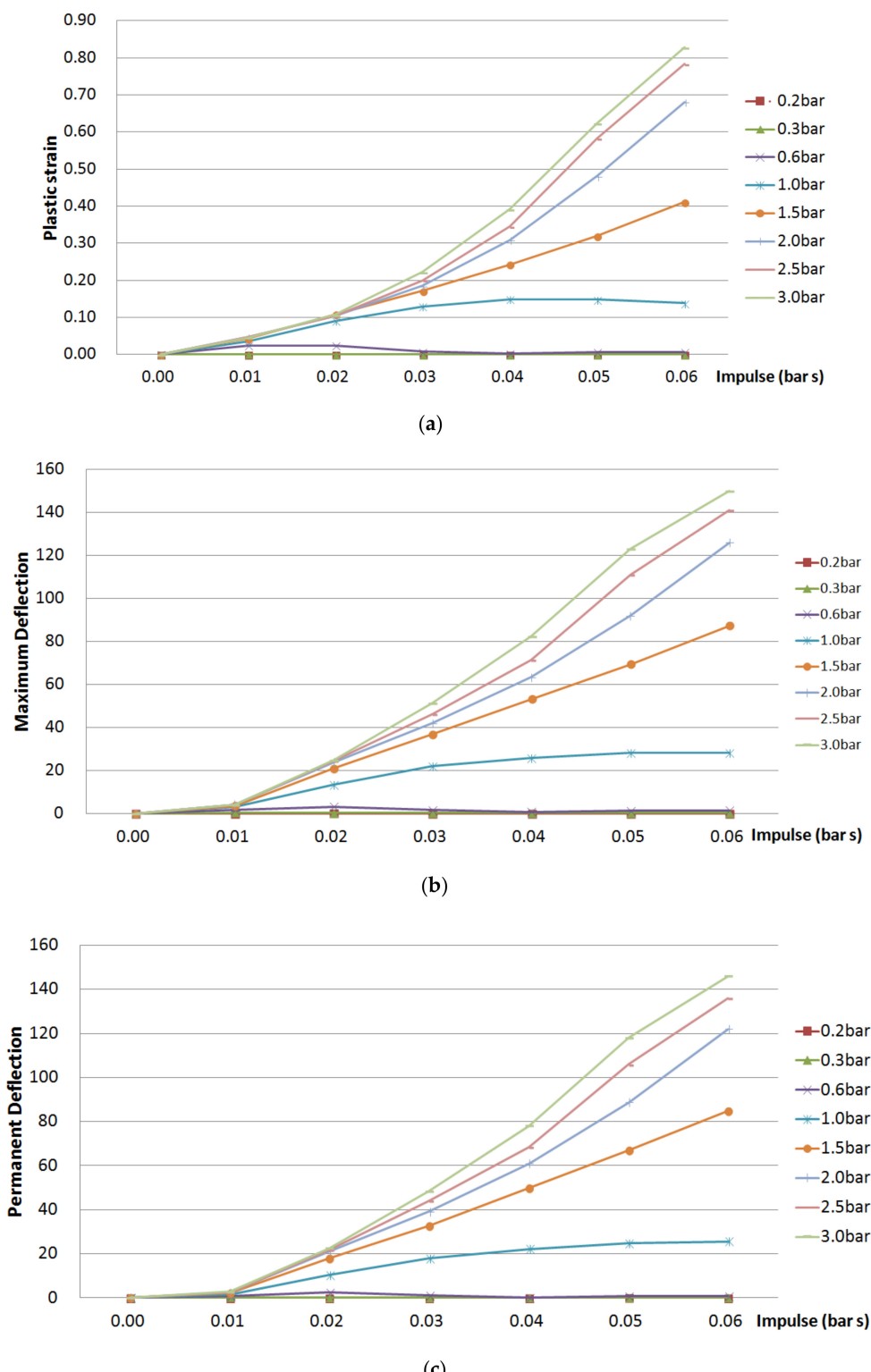

**Figure 10.** Relationship of structural response characteristics vs. impulse curve (at maximum point).
(**a**) Plastic strain—Impulse curve for each peak pressure. (**b**) Maximum deflection vs. impulse.
(**c**) Permanent deflection vs. impulse.

With peak pressure less than 1.0 bar, the permanent deflection was observed zero value, and with peak pressures greater than 1.0 bar, the maximum and permanent deflection are proportional to the peak pressure and impulse. However, in some ranges, the plastic strain and the maximum and

permanent deflection are not proportional to the peak pressure and impulse in the range of below 1.0 bar of peak pressure and impulse between 0.01 bar and 0.03 bar.

In Figure 11, the plastic strain results are zoomed and compared to the ranges of below 1.0 bar of peak pressure and impulse between 0.01 bar and 0.03 bar. One is the current actual thickness of 10mm in industrial application, and the other is the blast wall thickness of 2.0 mm plate. The results show that the plastic strain is not proportional to impulse in the actual (10-mm) thickness; however, it is proportional to impulse in the 2.0 mm thickness model. It is expected that there is an effect between the thick plate (vertical H-beam) and the thin plate (blast wall plate).

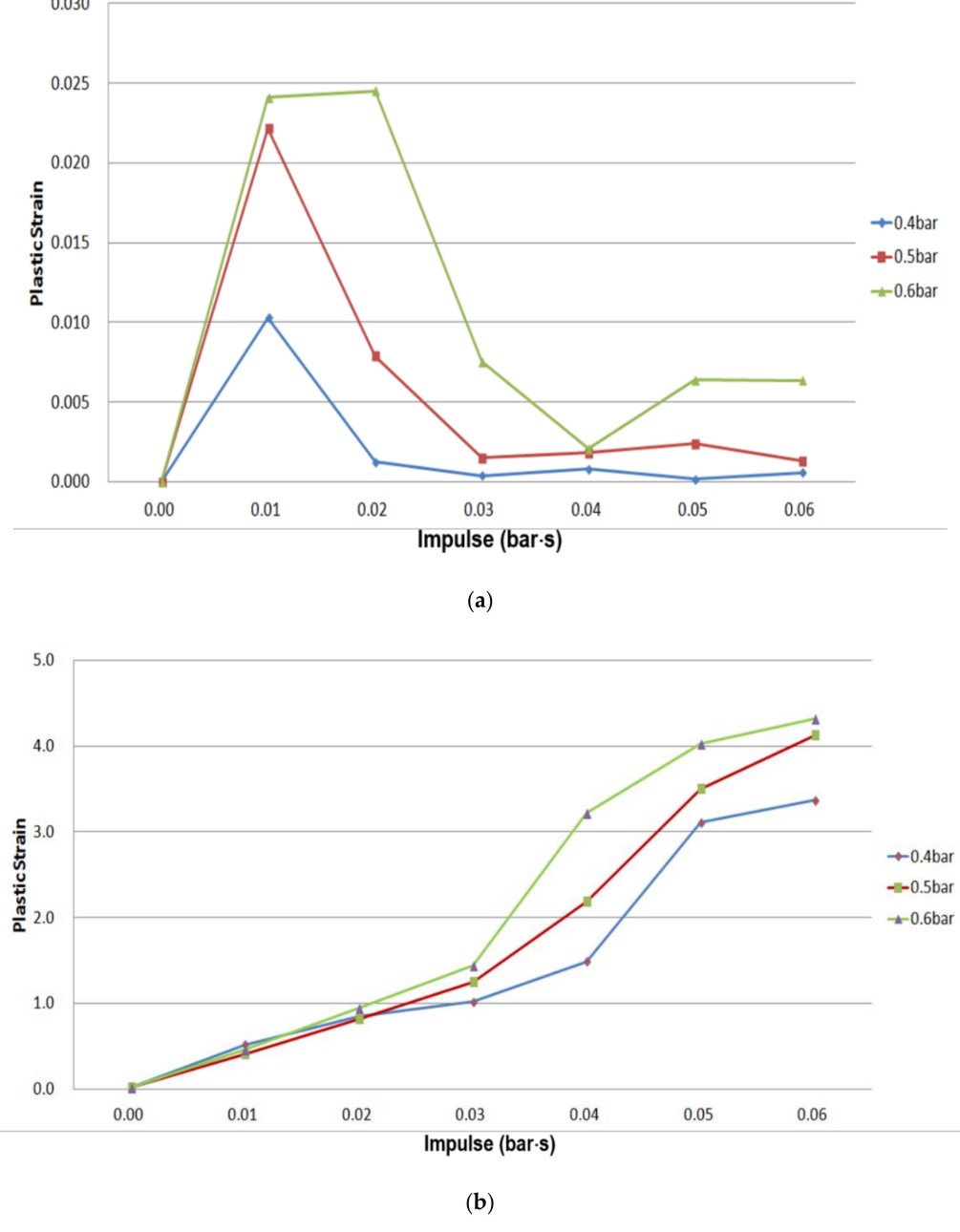

**Figure 11.** Plastic strain comparison. Actual plate thickness and thin plate. (**a**) Blast wall with actual plate thickness (**b**) Blast wall with 2.0.mm of plate thickness.

The maximum deflection is slightly higher than the permanent deflection, and the deflection with a peak pressure of greater than 1.0 bar is proportional to the peak pressure and impulse. However,

it also has an unusual range whose deflection is not proportional to the peak pressure and impulse. Figure 12 shows that the P-I curve for a previous study [10] and that for this study are very similar. This means that the elastic and plastic responses are nearly the same for the corrugated blast wall and the stiffened blast wall in this study when the same peak pressure and impulse are applied.

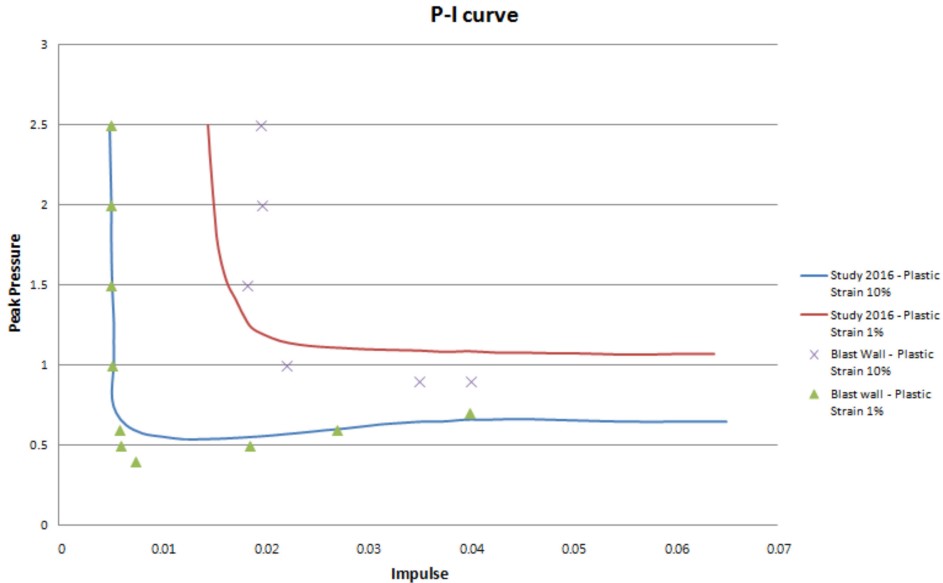

**Figure 12.** P-I curve comparison.

Table 3 shows the suggested critical strain [26] for various materials. In this study, the material of the blast wall was AH36, with a yield strength of 355 MPa and a plastic strain limit assumed to be 4%. Table 4 summarizes the resultant plastic strain according to the peak pressure and impulse. The under-bar values in the table are more than 4% of the plastic strain. Up to 0.6 bar, plastic strain did not occur at any location. With the peak pressure at 1.0 bar and above, the plastic strain was greater than 4% in most cases in which the impulse was higher than 0.1 bar and the higher peak pressure had a higher plastic strain.

**Table 3.** Maximum plastic strain, DNV-RP-C203 [26].

| | Maximum Plastic Strain | | | |
|---|---|---|---|---|
| **Critical Strain** | S235 0.05 | S355 0.04 | S420 0.03 | S460 0.03 |

**Table 4.** Resultant plastic strain for each peak pressure and impulse (at maximum point).

| Impulse (bar·s) | 0.2 bar | 0.3 bar | 0.6 bar | 1.0 bar | 1.5 bar | 2.0 bar | 2.5 bar | 3.0 bar |
|---|---|---|---|---|---|---|---|---|
| 0.0010 | 0.0000 | 0.0000 | 0.0000 | 0.0000 | 0.0000 | 0.0000 | 0.0000 | 0.0000 |
| 0.0100 | 0.0000 | 0.0014 | 0.0241 | 0.0350 | *0.0438* | *0.0466* | *0.0468* | *0.0461* |
| 0.0200 | 0.0000 | 0.0000 | 0.0245 | *0.0901* | *0.1085* | *0.1040* | *0.1043* | *0.1068* |
| 0.0300 | 0.0000 | 0.0000 | 0.0076 | *0.1298* | *0.1720* | *0.1850* | *0.2000* | *0.2230* |
| 0.0400 | 0.0000 | 0.0000 | 0.0021 | *0.1495* | *0.2430* | *0.3090* | *0.3450* | *0.3920* |
| 0.0500 | 0.0000 | 0.0000 | 0.0064 | *0.1481* | *0.3200* | *0.4820* | *0.5820* | *0.6230* |
| 0.0600 | 0.0000 | 0.0000 | 0.0064 | *0.1381* | *0.4110* | *0.6810* | *0.7830* | *0.8280* |

In the actual condition, as with the load profile [18], the negative phase is considered. To evaluate the effects of the negative phase of the explosion load, the load profile in Figure 13 has been simplified

and applied to the case with 0.6 bar in Table 4. In this case, 25% of the maximum pressure is applied for the minimum pressure, and the duration of the negative phase is twice that of the positive phase.

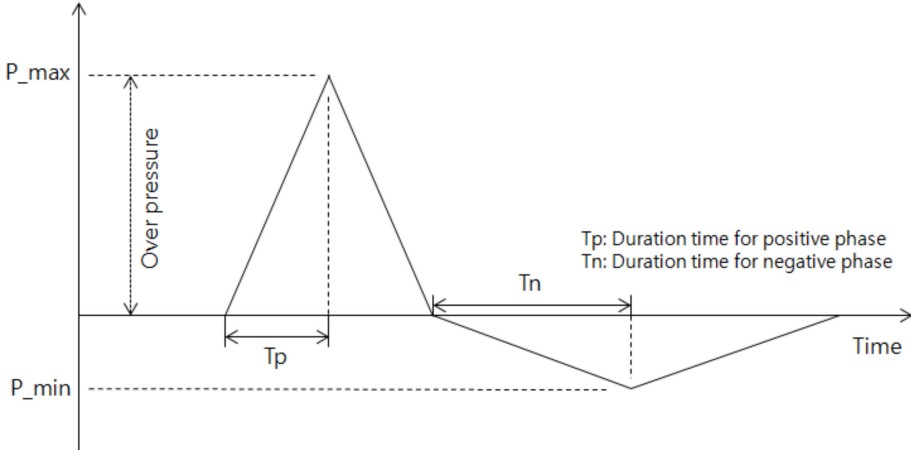

**Figure 13.** Idealized load profile with negative phase.

Figure 14 shows the plastic strain results: the plastic strain in the case with negative pressure is slightly greater than that in the case without negative pressure, and it is still below 5%. With these results, it can be concluded that the current blast wall structure has enough strength to endure the explosion pressure during the well test. The applied blast pressure in the calculation of the current design is 0.2 bar, but this study showed that the plastic strain begins to occur at 1.0 bar, which indicates a very low possibility that the blast wall will be deformed permanently because the explosion pressure 1.0 bar is not likely to occur. The high stresses in the blast wall structure are concentrated at the connection of the vertical H-beam and the boundary (deck plate), and the other locations experience very low stress relative to the end connection.

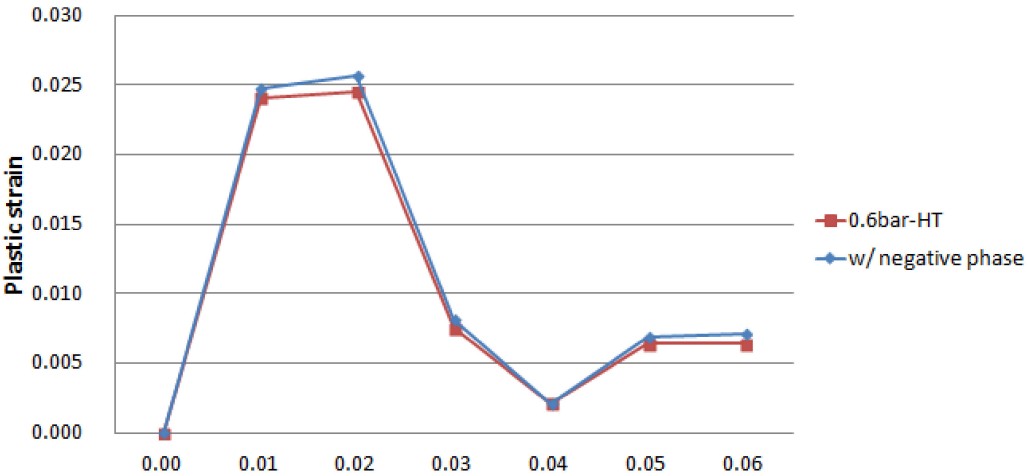

**Figure 14.** Plastic strain comparison with negative phase.

It can be proposed that this difference is because the current structure is evaluated following the linear beam theory, so the structure has a large vertical H-beam as a "primary" member, and the wall plates are connected to the H-beam as a "secondary" member. Therefore, the deformation of the blast wall is insufficient, and the explosion energy cannot be absorbed by the blast wall structure. A typical blast wall is thin corrugated steel panel because the blast pressure energy can be absorbed by the deformed corrugated panel. The deflection and plastic strain results for the stiffened panel blast wall within the area with permanent deformation are very similar to those of previous studies with a

corrugated blast wall. This finding indicates that the stiffened panel blast wall and the corrugated blast wall have similar structural dynamic responses.

## 3. Application to the Design of Blast Walls

The FE results from previous investigations show that the maximum plastic strain occurs at the bottom connection between the vertical girder (24-mm flange plate) and the blast wall plate (10-mm plate). Therefore, several alternative design applications are suggested to reduce the fabrication cost of a blast wall for application in current industrial practices. In this section, an applicable alternative design for existing blast wall is determined to reduce the plastic strain with following alternatives.

- Alt #1: Weld the blast wall plate and deck.
- Alt #2: Replace the blast wall with a thicker plate.
- Alt #3: Replace the blast wall and the support with mild steel.

With the three alternatives, the maximum von Mises stress and plastic strain are compared with the results of the current wall structure. Alt #1 and Alt #2 are compared with the result in Figures 15 and 16 (peak pressure: 1.5 bar; impulse: 0.02 bar), and Alt #3 is compared with the result in Figure 17 (peak pressure: 0.6 bar; impulse: 0.02 bar).

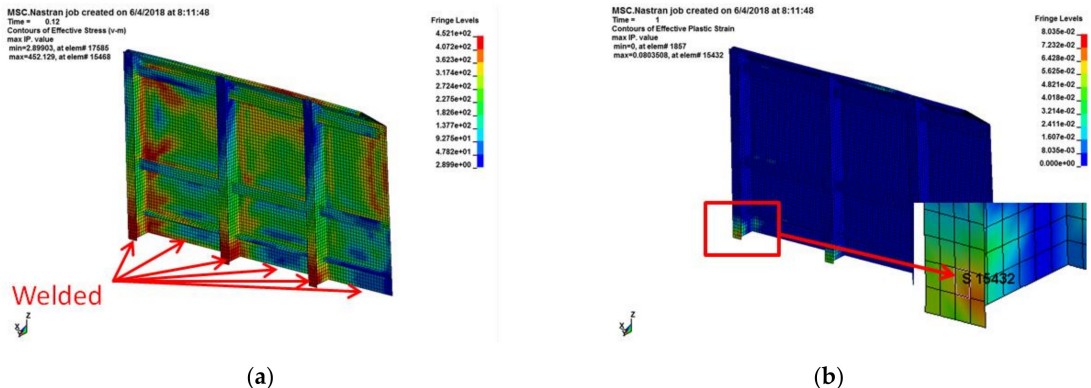

(a)            (b)

**Figure 15.** Results of alternative method Alt #1. (**a**) Updated boundary condition and stress distribution. (**b**) Strain distribution.

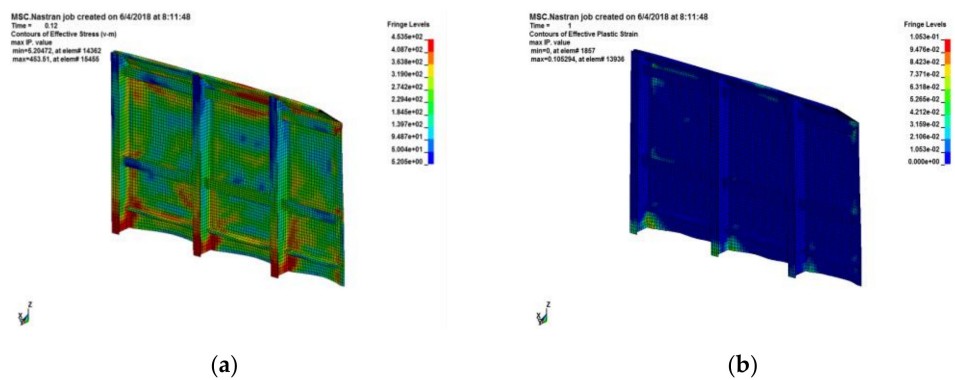

(a)            (b)

**Figure 16.** Updated blast wall plate thickness for wall plate thickness of 4 mm of Alt #2. (**a**) Von Mises stress distribution. (**b**) Strain distribution.

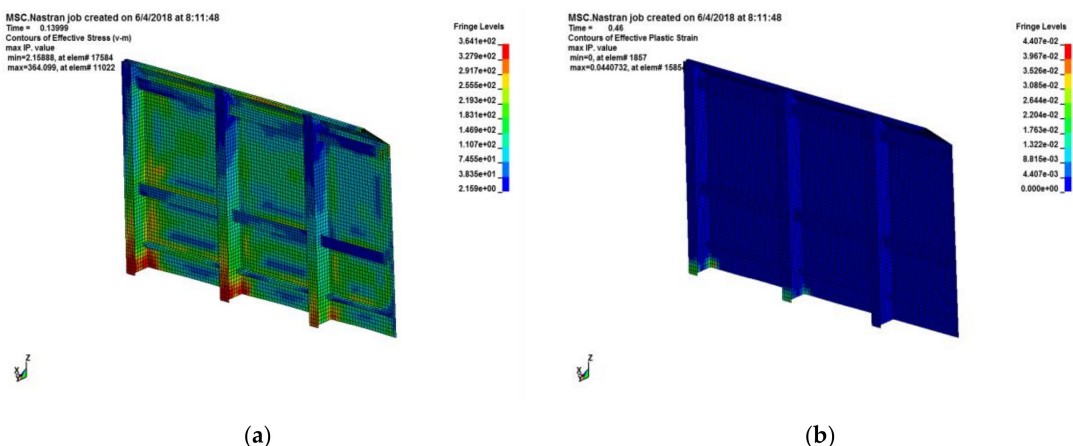

(**a**)                                                                 (**b**)

**Figure 17.** Maximum von Mises stress and plastic strain with wall mild steel of Alt #3. (**a**) Von Mises stress distribution. (**b**) Strain distribution.

The first alternative method is to change the welding conditions to weld the bottom of the blast wall plate and the existing welding stiffeners, as shown in Figure 15. Therefore, the location of the maximum plastic strain is changed from the plate to the stiffener. When the bottom of the blast wall plate is welded, the von Mises stress of the blast wall plate is greatly increased, but the maximum von Mises stress does not change. However, the plastic strain is more than 20% lower than in the original condition.

The second alternative method is to reduce the blast wall plate thickness (4.0 and 6.0 mm) to increase the absorbed energy. The maximum von Mises stress and plastic strain are shown with the different blast wall thicknesses. When the blast wall thickness is reduced, the von Mises stress and the plastic strain on the wall plate are increased.

The last alternative method is to use mild steel, which will reduce the cost of the current industrial project. Therefore, mild steel is used for the blast wall structure, and the results are compared with the current structure. To evaluate the plastic strain with the rule requirement, a case with a peak pressure of 0.6 bar and impulse of 0.02 bar is considered.

Figure 18 shows the plastic strain of the maximum point for the three cases such as original, Alt #1 (Full welding) and Alt #2 (P06, P15, P20, P04). The plastic strain is reduced in the case with a full welding boundary (Alt. #1), but the use of a thinner (or thicker) plate (Alt. #2) for the blast wall causes little change in plastic strain. This means that the blast wall thickness does not affect the plastic strain because the vertical supports are very rigid: in other words, the blast wall thickness can be reduced to the thinner plate.

Figure 19 shows the effects of different materials in the case of Alt. #3. The use of mild steel nearly doubled the plastic strain with high-tensile steel. Even though the plastic strain is increased in the case of mild steel, it is still below the rule (DNV-RP-C203) [26] requirement (5% in mild steel), which means that mild steel can be used for the blast wall. Therefore, it can be determined whether the current blast wall is over-designed due to blast loads.

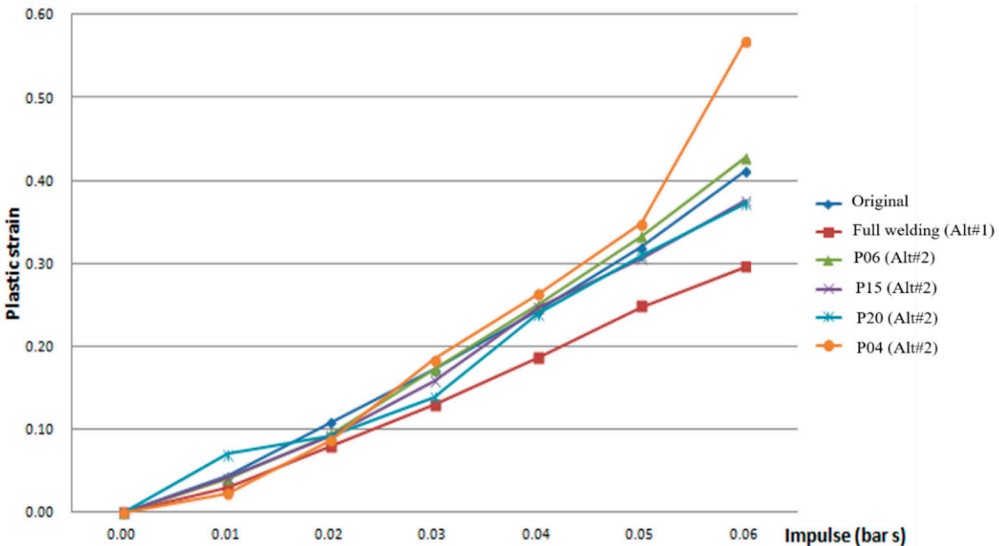

**Figure 18.** Plastic strain vs. impulse curves for Alt #1 and Alt #2 (at maximum strain point).

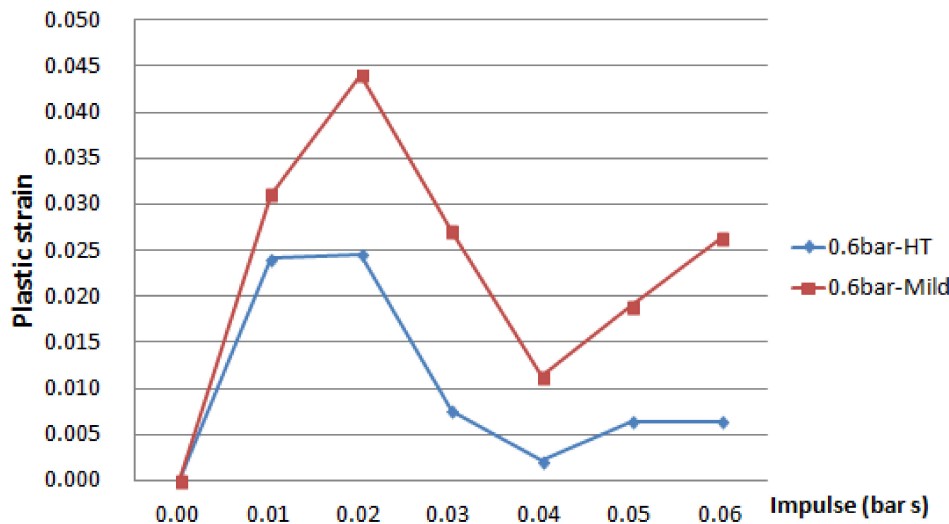

**Figure 19.** Plastic strain vs. impulse curve for Alt. #3.

## 4. Conclusions

The focus of this study is to determine whether the current scantling of the blast wall in a well-test area is adequate for the blast pressure. Three alternatives are also suggested to improve the strength characteristic of the blast wall and to reduce the cost of construction. The results of the analysis in this study lead us to the following conclusions.

The study result with the stiffened panel blast wall and the existing study results for a corrugated blast wall were very similar in deflection, plastic strain, and P-I curve. This means that the stiffened blast wall and the corrugated blast wall had very similar structural characteristics in the elastic and plastic regions and that developments can be made in the stiffened blast wall according to the corrugated blast wall study results.

The alternative with the additional weld between the blast wall and the deck has a great effect on reducing and delaying plastic strain. The blast wall thickness did not affect the plastic strain result because the vertical supports are very rigid, and the loads are all concentrated on the supports. This means that the thickness could be reduced if the vertical supports are retained in the blast wall structure. The blast wall could be changed from high-tensile steel to mild steel. The plastic strain

increases significantly with a mild steel structure, but it is still below the rule requirement. The cost of construction can be reduced with the use of mild steel. The results of the non-linear analysis in this study will be a good reference for future studies to apply a higher peak pressure than the current industrial practices.

**Author Contributions:** Conceptualization, B.J. and J.K.S.; Methodology, B.J. and J.K.S; Formal analysis, B.J. and J.H.K.; Investigation, B.J.; Data curation, B.J. and J.K.S.; Writing-original draft preparation, B.J.; Writing-review and editing, J.K.S.; Visualization, B.J., J.H.K., and J.K.S.; Supervision, J.K.S.; project administration, J.K.S.; funding acquisition, J.K.S. All authors have read and agreed to the published version of the manuscript.

**Funding:** This research was a part of the project titled "Development of guidance for prevent of leak and mitigation of consequence in hydrogen ships", funded by the Ministry of Oceans and Fisheries, Korea.

**Conflicts of Interest:** The authors declare no conflict of interest.

## Appendix A Load Scenario (Peak Pressure, 0.3 to 3.0 Bar)

**Table A1.** Load scenario (0.3 bar).

| Peak Pressure (bar) | Case Number | Duration Time (s) | Peak Time (s) | Impulse (bar) |
|---|---|---|---|---|
| | P03M01 | 0.0067 | 0.0033 | 0.001 |
| | P03M10 | 0.0667 | 0.0333 | 0.010 |
| | P03M20 | 0.1333 | 0.0667 | 0.020 |
| 0.3 | P03M30 | 0.2000 | 0.1000 | 0.030 |
| | P03M40 | 0.2667 | 0.1333 | 0.040 |
| | P03M50 | 0.3333 | 0.1667 | 0.050 |
| | P03M60 | 0.4000 | 0.2000 | 0.060 |

**Table A2.** Load scenario (0.6 bar).

| Peak Pressure (bar) | Case Number | Duration Time (s) | Peak Time (s) | Impulse (bar) |
|---|---|---|---|---|
| | P06M01 | 0.0033 | 0.0017 | 0.001 |
| | P06M10 | 0.0333 | 0.0167 | 0.010 |
| | P06M20 | 0.0667 | 0.0333 | 0.020 |
| 0.6 | P06M30 | 0.1000 | 0.0500 | 0.030 |
| | P06M40 | 0.1333 | 0.0667 | 0.040 |
| | P06M50 | 0.1667 | 0.0833 | 0.050 |
| | P06M60 | 0.2000 | 0.1000 | 0.060 |

**Table A3.** Load scenario (1.0 bar).

| Peak Pressure (bar) | Case Number | Duration Time (s) | Peak Time (s) | Impulse (bar) |
|---|---|---|---|---|
| | P10M01 | 0.0020 | 0.0010 | 0.001 |
| | P10M10 | 0.0200 | 0.0100 | 0.010 |
| | P10M20 | 0.0400 | 0.0200 | 0.020 |
| 1.0 | P10M30 | 0.0600 | 0.0300 | 0.030 |
| | P10M40 | 0.0800 | 0.0400 | 0.040 |
| | P10M50 | 0.1000 | 0.0500 | 0.050 |
| | P10M60 | 0.1200 | 0.0600 | 0.060 |

**Table A4.** Load scenario (1.5 bar).

| Peak Pressure (bar) | Case Number | Duration Time (s) | Peak Time (s) | Impulse (bar) |
|---|---|---|---|---|
| 1.5 | P15M01 | 0.0013 | 0.0007 | 0.001 |
| | P15M10 | 0.0133 | 0.0067 | 0.010 |
| | P15M20 | 0.0267 | 0.0133 | 0.020 |
| | P15M30 | 0.0400 | 0.0200 | 0.030 |
| | P15M40 | 0.0533 | 0.0267 | 0.040 |
| | P15M50 | 0.0667 | 0.0333 | 0.050 |
| | P15M60 | 0.0800 | 0.0400 | 0.060 |

**Table A5.** Load scenario (2.0 bar).

| Peak Pressure (bar) | Case Number | Duration Time (s) | Peak Time (s) | Impulse (bar) |
|---|---|---|---|---|
| 2.0 | P20M01 | 0.0010 | 0.0005 | 0.001 |
| | P20M10 | 0.0100 | 0.0050 | 0.010 |
| | P20M20 | 0.0200 | 0.0100 | 0.020 |
| | P20M30 | 0.0300 | 0.0150 | 0.030 |
| | P20M40 | 0.0400 | 0.0200 | 0.040 |
| | P20M50 | 0.0500 | 0.0250 | 0.050 |
| | P20M60 | 0.0600 | 0.0300 | 0.060 |

**Table A6.** Load scenario (2.5 bar).

| Peak Pressure (bar) | Case Number | Duration Time (s) | Peak Time (s) | Impulse (bar) |
|---|---|---|---|---|
| 2.5 | P25M01 | 0.0008 | 0.0004 | 0.001 |
| | P25M10 | 0.0080 | 0.0040 | 0.010 |
| | P25M20 | 0.0160 | 0.0080 | 0.020 |
| | P25M30 | 0.0240 | 0.0120 | 0.030 |
| | P25M40 | 0.0320 | 0.0160 | 0.040 |
| | P25M50 | 0.0400 | 0.0200 | 0.050 |
| | P25M60 | 0.0480 | 0.0240 | 0.060 |

**Table A7.** Load scenario (3.0 bar).

| Peak Pressure (bar) | Case Number | Duration Time (s) | Peak Time (s) | Impulse (bar) |
|---|---|---|---|---|
| 3.0 | P30M01 | 0.0007 | 0.0003 | 0.001 |
| | P30M10 | 0.0067 | 0.0033 | 0.010 |
| | P30M20 | 0.0133 | 0.0067 | 0.020 |
| | P30M30 | 0.0200 | 0.0100 | 0.030 |
| | P30M40 | 0.0267 | 0.0133 | 0.040 |
| | P30M50 | 0.0333 | 0.0167 | 0.050 |
| | P30M60 | 0.0400 | 0.0200 | 0.060 |

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
