# Peer review of "Investigation of the Structural Strength of Existing Blast Walls in Well-Test Areas on Drillships"

_jmse, doi:10.3390/jmse8080583_

Round 1
Reviewer 1 Report
An interesting article that the results can be used in practical solutions.
The article has minor editorial shortcomings:
- The descriptions in Figure 2 are difficult to read
- Section 2.1, I would start with the text from verse 110 and only later would I describe figures 2 and 3
- The descriptions in Figure 8 are difficult to read
Author Response
Reviewer #1
An interesting article that the results can be used in practical solutions. The article has minor editorial shortcomings:
[Reply] The authors greatly appreciate the reviewer’s comments and suggestions for revising the paper.
[1] The descriptions in Figure 2 are difficult to read
[Reply] Thank you for this comment. Fig. 2 has been modified for reading properly
[2] Section 2.1, I would start with the text from verse 110 and only later would I describe figures 2 and 3
[Reply] Thank you for this comment. Section 2.1 has been modified as per reviewer’s suggestions.
[3] The descriptions in Figure 8 are difficult to read
[Reply] Thank you for this comment. Fig. 8 has been modified for reading properly

Reviewer 2 Report
The manuscript describes an interesting analysis of blast loading of the reinforced protective walls currently in use on the drill ships. The methodology is based on a simulation programme conducted in LS-Dyna and reported adequately. The manuscript is structured well and should be accepted for publication, subject to the list of corrections given below:
- More specific outcomes and key results of this work to be provided in the Abstract (at the end of);
- Reference for corrugated solutions can be added to [4] from elsewhere, for instance https://doi.org/10.1016/j.tws.2019.106277 ;
- For shock loading analysis a number of references can be added to reference to LS-Dyna [10], including https://doi.org/10.1016/j.ijplas.2018.02.014, https://doi.org/10.1016/j.ijimpeng.2014.11.002
- Publication outline to be added at the end of Introduction for the sake of presentation clarity;
- Text in Figure 2 not very clear/readable – should be either deleted or replaced in a tidier format;
- Annual rate of the well tests being 30 to 45 repeated several times in the main body of the text;
- Five scenarios described in line 132 – is the quoted number of pipes per side or total?
- Figure 5 – what exactly are part a) and b) and state clearly where the measurements were taken;
- It needs to be stated clearly if four categories classification given in Figure 6 are authors approach or taken from the literature, the latter needs to be supported with an appropriate reference;
- Discussion below Figure 6 can be improved and expanded as it is not clear what evidence for the statements provided is;
- It is not clear why the authors used symmetric triangular loading and how does it compare with the recording shown in Figure 5;
- Check the Equation (2) and data presented in Table 2;
- It is not clear how is the yield stress converted to true stress (see line 254);
- Check the results in Figure 12 for the maximum deflection and the low levels of pressure – for the first couple of curves maximum deflection seem to be zero for impulse between 0.05 and 0.06;
- Results in Figure 13 are not clear – what is the horizontal axis;
- Figure 20 is missing;
- Discussion about the results shown in Figure 21 is not consistent with the legend used in the graph;
- Authors should provide clarification on the plastic strain of 4% and rule requirement of 5% quoted near line 391 (the latter should be supported with appropriate book of requirements);
- It is not clear if the authors use MSc Nastran (with appear in the heading of the results plot) or LS Dyna in the simulation programme;
Author Response
The manuscript describes an interesting analysis of blast loading of the reinforced protective walls currently in use on the drill ships. The methodology is based on a simulation programme conducted in LS-Dyna and reported adequately. The manuscript is structured well and should be accepted for publication, subject to the list of corrections given below:
[Reply] The authors greatly appreciate the reviewer’s comments, suggestions, and positive opinion for revising the paper.
[1] More specific outcomes and key results of this work to be provided in the Abstract (at the end of);
[Reply] Thank you for your suggestion. Author has been modified the abstract for Key outcomes and results as below
“Blast walls are installed on the topside of offshore structures to reduce the damage from fire and explosion accidents. The blast walls on production platforms such as floating production storage, offloading, and floating production units undergo fire and explosion risk analysis, but information about blast walls on the well test area of drillship topsides is insufficient even though well tests are performed 30 to 45 times per year. Moreover, current industrial practices of design method are used as a simplified elastically design approaches. Therefore, this study investigates the strength characteristic of blast wall on drillship based on the blast load profile from fire and explosion risk analysis results, as well as the ability of the current design scantling of the blast wall to endure the blast pressure during the well test. The maximum plastic strain of the FE results occurs at the bottom connection between the vertical girder and the blast wall plate. Based on the results, several alternative design applications are suggested to reduce the fabrication cost of a blast wall such as differences of stiffened plated structure and corrugated panels, possibility of changing material (mild steel), reduced plate thickness, for application in current industrial practices.”
[2] Reference for corrugated solutions can be added to [4] from elsewhere, for instance https://doi.org/10.1016/j.tws.2019.106277 ;
[Reply] Thank you for this comment. Author added the reference paper in manuscript.
[5]Vignjevic, R.; Liang, C.; Hughes, K.; Brown, J.C.; Vuyst, T.D.; Djordjevic, N.; Campbell, J. A numerical study on the influence of internal corrugated reinforcements on the biaxial bending collapse of thin-walled beams. Thin-Walled Structures. 2019, 106277. https://doi.org/10.1016/j.tws.2019.106277
[3] For shock loading analysis a number of references can be added to reference to LS-Dyna [10], including https://doi.org/10.1016/j.ijplas.2018.02.014, https://doi.org/10.1016/j.ijimpeng.2014.11.002
[Reply] Thank you for this comment. Author added the reference paper in manuscript.
[12]Djordjevic, N.; Vignjevic, R.; Kiely, L.; Case, S.; Vuyst, T.D.; Campbell, J.; Hughes, K. Modelling of shock waves in fcc and bcc metals using a combined continuum and dislocation kinetic approach. International Journal of Plasticity. 2018, 105, 211-224.
[13]Vignjevic, R.; Hughes, K.; Vuyst, T.D.; Djordjevic, N.; Campbell, J.; Stojkovic, M.; Gulavani, O.; Hiermaier, S. Lagrangian analysis led design of a shock recovery plate impact experiment. International Journal of Impact Engineering. 2015, 77, 16-29.
[4] Publication outline to be added at the end of Introduction for the sake of presentation clarity;
[Reply] Thank you for your comments. As per the reviewer’s comment, the authors have added at the end of the introduction.
“According to existing blast wall studies are well developed, validated and suggested. However, limited studies have examined the blast wall in the well test area on the topside of a drillship, likely because although floating offshore platforms (e.g., floating liquid natural gas and floating production storage and offloading units) have great exposure to the risk of fire and explosion accidents, accidents aboard drillships are expected to occur during well test operation.”
[5] Text in Figure 2 not very clear/readable – should be either deleted or replaced in a tidier format;
[Reply] Thank you for this comment. Fig. 2 has been modified for reading properly
[6] Annual rate of the well tests being 30 to 45 repeated several times in the main body of the text;
[Reply] Thank you for this comments. As per the reviewer’s comment, the authors have modified and repeating annual rate expression.
[7] Five scenarios described in line 132 – is the quoted number of pipes per side or total?
[Reply] The quoted number of pipes per total is provide the experimental study. The pipe rack structure was designed with 48 pipes (Fig. 4b). The details of information given in the referred paper.
[15]Bae, M.H.; Paik, J.K. Effects of structural congestion and surrounding obstacles on the overpressure loads in explosions: Experiment and CFD simulations. Ships and Offshore Structures, 2018, 13(2), 165-180.
[8] Figure 5 – what exactly are part a) and b) and state clearly where the measurements were taken;
[Reply] Thank you for this comment. Figs. 5(a) and (b) show the typical overpressure profile of experimental results such as load vs. time. The measurements points are indicated at MP 1, 11 as shown in below Figure. 9 which is referred in [15]. Bae and Paik (2018) are measured at various point (surrounded pipe rack). The overpressure loads and related pressure characteristics with time were measured using pressure sensors attached to the points of interest in the test module. Twenty-four pressure sensors were installed where the explosion loads were sensitively affected. As per reviewer's comment, authors believe that detail of experimental setup can be taken from reference paper in this paper.
Figure 9. Locations of the pressure sensors. (a) Elevation view of pressure sensors; (b) locations of pressure sensors. (This figure is available in colour online).[15]
[15]Bae, M.H.; Paik, J.K. Effects of structural congestion and surrounding obstacles on the overpressure loads in explosions: Experiment and CFD simulations. Ships and Offshore Structures, 2018, 13(2), 165-180.
[9] It needs to be stated clearly if four categories classification given in Figure 6 are authors approach or taken from the literature, the latter needs to be supported with an appropriate reference;
[Reply] Thank you for this comment. As per figure 6, 4 types of loading shapes are provided with an appropriate references as below
[9] Sohn and Kim (2017) provide details of the loading shapes and added original reference [16] Biggs, J.M. Introduction to Structural Dynamics, McGraw-Hill Inc.1964. (New York).
[10] Discussion below Figure 6 can be improved and expanded as it is not clear what evidence for the statements provided is;
[Reply] As per reviewers’s comments, the discussion has been modified.
[11] It is not clear why the authors used symmetric triangular loading and how does it compare with the recording shown in Figure 5;
[Reply] Thanks for reviewer's comment. For selection of blast load shape to the blast wall of drillship, it is important loading shape properly. However, previous researchers have been fully investigated and discussed the applicable loading shapes [Sohn and Kim (2017)]. Therefore, this study does not have to be re-produced for tendency studies of loading shape. As per previous results and discussion was conducted and selected the applied symmetric triangular loading shape in this study.
[12] Check the Equation (2) and data presented in Table 2;
[Reply] Thank for comment. Expression of impulse is area of time and pressure. The equation (2) is able to be expressed as a triangular area which is impulses.
[13] It is not clear how is the yield stress converted to true stress (see line 254);
[Reply] Thanks for reviewer comment. The expression was wrong. Yield stress is able to be converted to dynamic yield stress depend on strain rate. This expression has been modified.
[14] Check the results in Figure 12 for the maximum deflection and the low levels of pressure – for the first couple of curves maximum deflection seem to be zero for impulse between 0.05 and 0.06;
[Reply] Thanks for comment. Impulse ranges between 0.05 and 0.06 were calculated almost zero value (Not zero value) which is able to explain that low level of pressure tends to be not effected structural and material behaviour at maximum point of the target structure.
[15] Results in Figure 13 are not clear – what is the horizontal axis;
Figure 20 is missing;
[Reply] Figure 12 has been modified as the horizontal axis which is impulse.
[16] Discussion about the results shown in Figure 21 is not consistent with the legend used in the graph;
[Reply] Thanks for this comments. We revised the Figure 21 modified the legend with discussion part.
[17] Authors should provide clarification on the plastic strain of 4% and rule requirement of 5% quoted near line 391 (the latter should be supported with appropriate book of requirements);
[Reply] thanks for this comment. Authors provide clarification rules of plastic stain ranges for design approach.
[18] It is not clear if the authors use MSc Nastran (with appear in the heading of the results plot) or LS Dyna in the simulation programme;
[Reply] Thanks for comment. A commercial FE code (MSC Nastran) was employed in this study. Author add the information in the manuscript and reference.

Reviewer 3 Report
The authors have done a wonderful study for explaining how blast walls on the topside of offshore structures can reduce the damage from fire and explosion accidents. I think the study presented here has novelty but there is a lack of thorough comparison of literature. I think authors have to thoroughly compare their work with other studies, I have given some suggestions below.
Overall, the manuscript is well written. I would accept the paper after minor revision for answering my questions and as well as proposed citations. Carefully reading and answering all comments will raise the article for more readership.
- Please provide the abstract in below way. I think some information is missing. Please provide a Hypothesis first of your work. Then provide an experimental procedure and then discuss your findings. For now, the abstract is not in proper format.
- Please also provide alternative drilling strategies in the Introduction section. You can also discuss that drilling fluids used in drilling operations can also reduce the risk of fire and blast, for example, water-based drilling fluids (WBDF), which, if used with proper additives like nano-particles can be more effective than conventional oil-based drilling fluids and also can mitigate the firing problems for high pressure and high-temperature wells. Oil-based drilling fluids can be toxic to the marine species, if any incident of oil spills happens, like the Gulf of Mexico, whereas, suggested WBDF with nano-particles are non-toxic. There are few studies on this. Please also cite them, as below:
Ali, M., Jarni, H. H., Aftab, A., Ismail, A. R., Saady, N. M. C., Sahito, M. F., ... & Sarmadivaleh, M. (2020). Nanomaterial-Based Drilling Fluids for Exploitation of Unconventional Reservoirs: A Review. Energies, 13(13), 3417.
Aftab, A.; Ali, M.; Sahito, M.F.; Mohanty, U.S.; Jha, N.K.; Akhondzadeh, H.; Azhar, M.R.; Ismail, A.R.; Keshavarz, A.; Iglauer, S. Environmental-friendly and high-performance of multifunctional Tween 80/ZnO nanoparticles added water-based drilling fluid: an experimental approach. Sustain. Chem. Eng. 2020, accepted.
Aftab, A., Ali, Arif, M., Al-khdeawee, E., Saady, N. M. C., Ismail, A. R., M., Sahito, M. F., Keshavarz, A., Iglauer, S. Influence of Tailored-Made TiO2/API Bentonite Nanocomposite in Drilling Mud; Implications for Enhanced Drilling Operations. Applied Clay Science, 2020, accepted.
- Please provide a broad comparison between previous studies from literature and your study in the discussion section.
- Figure 2 is blurry, please provide a clear figure. Similarly, figure 8 is blurry, please provide a clear figure.
- Please check the spelling and grammatical error, I have seen some while reading. It requires minor English editing.
- The paper is written well, but somehow, the reader loses their interest, please shorten the sections, some of them are in too much detail and there are some repetitions of the work.
- There are many places where the hypothesis is said, but without the references, especially in the introduction and discussion section. Please provide the citations of the relevant work based on your hypothesis.
Hope this helps.
Ali
Author Response
Reviewer #3
The authors have done a wonderful study for explaining how blast walls on the topside of offshore structures can reduce the damage from fire and explosion accidents. I think the study presented here has novelty but there is a lack of thorough comparison of literature. I think authors have to thoroughly compare their work with other studies, I have given some suggestions below.
Overall, the manuscript is well written. I would accept the paper after minor revision for answering my questions and as well as proposed citations. Carefully reading and answering all comments will raise the article for more readership.
[Reply] Thank you for the valuable comments and positive opinion. Authors are revised as per reviewer’s comments.
[1] Please provide the abstract in below way. I think some information is missing. Please provide a Hypothesis first of your work. Then provide an experimental procedure and then discuss your findings. For now, the abstract is not in proper format.
[Reply] Thank you for your suggestion. Author has been modified the abstract for Key outcomes and results as below.
“Blast walls are installed on the topside of offshore structures to reduce the damage from fire and explosion accidents. The blast walls on production platforms such as floating production storage, offloading, and floating production units undergo fire and explosion risk analysis, but information about blast walls on the well test area of drillship topsides is insufficient even though well tests are performed 30 to 45 times per year. Moreover, current industrial practices of design method are used as a simplified elastically design approaches. Therefore, this study investigates the strength characteristic of blast wall on drillship based on the blast load profile from fire and explosion risk analysis results, as well as the ability of the current design scantling of the blast wall to endure the blast pressure during the well test. The maximum plastic strain of the FE results occurs at the bottom connection between the vertical girder and the blast wall plate. Based on the results, several alternative design applications are suggested to reduce the fabrication cost of a blast wall such as differences of stiffened plated structure and corrugated panels, possibility of changing material (mild steel), reduced plate thickness, for application in current industrial practices.”
[2]Please also provide alternative drilling strategies in the Introduction section. You can also discuss that drilling fluids used in drilling operations can also reduce the risk of fire and blast, for example, water-based drilling fluids (WBDF), which, if used with proper additives like nano-particles can be more effective than conventional oil-based drilling fluids and also can mitigate the firing problems for high pressure and high-temperature wells. Oil-based drilling fluids can be toxic to the marine species, if any incident of oil spills happens, like the Gulf of Mexico, whereas, suggested WBDF with nano-particles are non-toxic. There are few studies on this. Please also cite them, as below:
Ali, M., Jarni, H. H., Aftab, A., Ismail, A. R., Saady, N. M. C., Sahito, M. F., ... & Sarmadivaleh, M. (2020). Nanomaterial-Based Drilling Fluids for Exploitation of Unconventional Reservoirs: A Review. Energies, 13(13), 3417.
Aftab, A.; Ali, M.; Sahito, M.F.; Mohanty, U.S.; Jha, N.K.; Akhondzadeh, H.; Azhar, M.R.; Ismail, A.R.; Keshavarz, A.; Iglauer, S. Environmental-friendly and high-performance of multifunctional Tween 80/ZnO nanoparticles added water-based drilling fluid: an experimental approach. Sustain. Chem. Eng. 2020, accepted.
Aftab, A., Ali, Arif, M., Al-khdeawee, E., Saady, N. M. C., Ismail, A. R., M., Sahito, M. F., Keshavarz, A., Iglauer, S. Influence of Tailored-Made TiO2/API Bentonite Nanocomposite in Drilling Mud; Implications for Enhanced Drilling Operations. Applied Clay Science, 2020, accepted.
[Reply] Thank you for this comment. Author added the reference papers in manuscript.
[3]Please provide a broad comparison between previous studies from literature and your study in the discussion section.
[Reply] Thank you for this comment. Author provided a broad comparison between previous studies in end of introduction.
[4]Figure 2 is blurry, please provide a clear figure. Similarly, figure 8 is blurry, please provide a clear figure.
[Reply] Thank you for this comment. Fig. 2 has been modified for reading properly
[5]Please check the spelling and grammatical error, I have seen some while reading. It requires minor English editing.
[Reply] Thank you for your comment. The paper has been revised and edited by a native speaker.
[6]The paper is written well, but somehow, the reader loses their interest, please shorten the sections, some of them are in too much detail and there are some repetitions of the work.
[Reply] Thank you for this comments. As per the reviewer’s comment, the authors have modified and repeating annual rate expression.
[7]There are many places where the hypothesis is said, but without the references, especially in the introduction and discussion section. Please provide the citations of the relevant work based on your hypothesis.
[Reply] Thank you for this comments. As per the reviewer’s comment, the authors have been modified whole parts with references.
